# Size-Based Effects of Anthropogenic Ultrafine Particles on Lysosomal TRPML1 Channel and Autophagy in Motoneuron-like Cells

**DOI:** 10.3390/ijms232113041

**Published:** 2022-10-27

**Authors:** Silvia Sapienza, Valentina Tedeschi, Barbara Apicella, Francesco Palestra, Carmela Russo, Ilaria Piccialli, Anna Pannaccione, Stefania Loffredo, Agnese Secondo

**Affiliations:** 1Department of Neuroscience, Reproductive and Odontostomatological Sciences, University of Naples Federico II, 80131 Naples, Italy; 2Istituto di Scienze e Tecnologie per l’Energia e la Mobilità Sostenibili (STEMS)-CNR, 80125 Naples, Italy; 3Department of Translational Medical Sciences, Center for Basic and Clinical Immunology Research (CISI), WAO Center of Excellence, University of Naples Federico II, 80131 Naples, Italy; 4Institute of Experimental Endocrinology and Oncology (IEOS), National Research Council, 80131 Naples, Italy

**Keywords:** ultrafine particles, air pollution, TRPML1, autophagy, motor neurons, neurotoxicity, lysosome, mitochondrial dysfunction, ROS

## Abstract

Background: An emerging body of evidence indicates an association between anthropogenic particulate matter (PM) and neurodegeneration. Although the historical focus of PM toxicity has been on the cardiopulmonary system, ultrafine PM particles can also exert detrimental effects in the brain. However, only a few studies are available on the harmful interaction between PM and CNS and on the putative pathomechanisms. Methods: Ultrafine PM particles with a diameter < 0.1 μm (PM0.1) and nanoparticles < 20 nm (NP20) were sampled in a lab-scale combustion system. Their effect on cell tracking in the space was studied by time-lapse and high-content microscopy in NSC-34 motor neurons while pHrodo™ Green conjugates were used to detect PM endocytosis. Western blotting analysis was used to quantify protein expression of lysosomal channels (i.e., TRPML1 and TPC2) and autophagy markers. Current-clamp electrophysiology and Fura2-video imaging techniques were used to measure membrane potential, intracellular Ca^2+^ homeostasis and TRPML1 activity in NSC-34 cells exposed to PM0.1 and NP20. Results: NP20, but not PM0.1, reduced NSC-34 motor neuron movement in the space. Furthermore, NP20 was able to shift membrane potential of motor neurons toward more depolarizing values. PM0.1 and NP20 were able to enter into the cells by endocytosis and exerted mitochondrial toxicity with the consequent stimulation of ROS production. This latter event was sufficient to determine the hyperactivation of the lysosomal channel TRPML1. Consequently, both LC3-II and p62 protein expression increased after 48 h of exposure together with AMPK activation, suggesting an engulfment of autophagy. The antioxidant molecule Trolox restored TRPML1 function and autophagy. Conclusions: Restoring TRPML1 function by an antioxidant agent may be considered a protective mechanism able to reestablish autophagy flux in motor neurons exposed to nanoparticles.

## 1. Introduction

Atmospheric particulate matter (PM) is recognized as a major air pollutant and one of the main contributors to the global burden of disease [1]. An emerging body of evidence indicates an association between particulate air pollution and neurodegeneration [2,3], highlighting oxidative stress as one of the most important pathomechanisms [3,4,5,6,7].

PM of anthropogenic origin is mainly due to motor vehicle emissions and industrial combustion, covering almost all the fine (PM2.5, PM < 2.5 μm) up to ultrafine (PM0.1, PM < 0.1 μm) particulate matter fractions [8]. Alarmingly, PM0.1 levels are unmonitored and, therefore, still unregulated due to the limited detection capability of the diagnostic techniques currently used [9], even if the relative exposure is estimated to be very high [10]. Poor knowledge is especially present for PM0.1, whose presence has been observed in human olfactory bulb periglomerular neurons and in intraluminal erythrocytes of frontal lobe and trigeminal ganglia capillaries [11,12,13]. Analysis of brain tissue from individuals residing in highly polluted areas shows an increase in infiltrating monocytes or resident microglia activation and in detrimental protein aggregation together with blood–brain barrier (BBB) disruption [14]. Of note, prefrontal lobe lesions and neurochemical/behavioral changes have also been described in patients chronically exposed to air pollution [15,16,17,18,19,20,21].

Molecularly, PM0.1 and, in particular, its sub-20 nm fraction (the so-called nanoparticles, NPs) can enter the brain through active transport, blood–brain barrier leak, and passage into the olfactory bulb [12], thus triggering neurodegenerative diseases [14,22,23,24,25]. Accordingly, it has been found in both human brain capillaries and parenchyma [11], suggesting its ability to cross the BBB [26]. In fact, from the olfactory bulb, inhaled nanoparticles reach trigeminal nerves, brainstem, and hippocampus [13,27], entering deep into the brain parenchyma [15]. This passage determines a marked decrease in cognitive function in humans [11,12]. Of note, new technology combustion systems determine the emission of those smallest particles into the atmosphere. However, NPs’ biological activity remains partially unknown, possibly due to the difficulties in measuring their level at the exhausts and in the atmosphere. Unfortunately, this limitation has prevented over time an exhaustive study of the biological activity of these relevant air pollution constituents. Moreover, atmospheric PMs are conjugated with organic carbon (OC)-forming complex particles with a variable chemical composition that may influence their cellular effects [28,29]. OC has been characterized in detail in several previous works ([30] and references therein), and it was found to be constituted of polycyclic aromatic hydrocarbons (PAHs) and high-molecular-weight aromatic species. By contrast, the PM0.1 were found to be constituted almost completely by carbon with a low percentage of hydrogen (typically H/C is around 0.1 ([31] and references therein). In this study, PM0.1 were deprived of OC fraction in order to establish their size-dependent effect on lysosomal function in motor neurons. Particles were also further separated by a properly developed analytical protocol for isolating the nanoparticles < 20 nm (NP20).

Beyond their canonical role in digesting intracellular biomolecules and dysfunctional organelles, lysosome seems to precede tissue homeostasis, cell death and survival, nutrient sensing, autophagy, and Ca^2+^ ion storing and handling [32]. Accordingly, lysosomes are now considered as a quick exchange Ca^2+^ store equipped with their own functional apparatus formed by ion channels and pumps by which they control local [32] and cytosolic Ca^2+^ homeostasis [33]. Interestingly, dysfunction of lysosomal channels may underlie the pathogenesis of many lysosomal storage diseases, other metabolic disorders, and some neurological diseases [33,34,35,36,37,38]. In line with this view, we have previously demonstrated the involvement of the lysosomal cationic channel TRPML1 (also called mucolipin-1) and lysosomal calcium dyshomeostasis in the pathogenesis of amyotrophic lateral sclerosis/Parkinson–dementia complex [34,36,37]. Furthermore, a reactive oxygen species (ROS)-dependent hyperactivation of TRPML1 may lead to detrimental effects in neurons exposed to low oxygen conditions mimicking brain ischemia [35]. Moreover, TRPML1 activation may trigger autophagy through TFEB nuclear translocation [39]. In consideration of these findings, in the present study, the putative involvement of lysosome-dependent autophagy in size-based effects of anthropogenic ultrafine particles has been studied. To this aim, the correlation between oxidative stress and TRPML1 dysfunction has been investigated in vitro as a detrimental signaling pathway in motor-neuronal cells exposed to nanoparticles derived from air pollution. This event may represent a putative mechanism of air-pollutant nanoparticles underlying neurodegeneration in humans.

## 2. Results

### 2.1. Effects of PM0.1 and NP20 on Cell Tracking and Membrane Potential of NSC-34 Motor Neurons

Since different stimuli can affect cell movement in space, NSC-34 motor neuron tracking was investigated with time-lapse and high-content microscopy Operetta High-Content Imaging System (PerkinElmer). To characterize the impact of PM0.1 and NP20 on cell movement, NSC-34 motor neurons were exposed to complete medium alone, PM0.1 or NP20 over 6 h at 37 °C. Setting the start point of each cell at 0 on the X-axis and 0 on the Y-axis, NP20 induced changes, starting from 50 min, in cell displacement along the X-axis measured as current displacement X (Figure 1A(a)) and displacement X mean per well (Figure 1A(b)), but did not affect movement on the Y-axis (Appendix A). In particular, NP20-exposed NSC-34 motor neurons displayed less distance from point of origin (Figure 1A(a,b)) compared to untreated cells. Conversely, the cellular speed between NP20-untreated and -treated NSC-34 motor neurons was similar (Figure 1B). PM0.1, unlike NP20, did not affect NSC-34 cell tracking compared to the control. Furthermore, current-clamp experiments revealed that NP20, but not PM0.1, was able to shift membrane potential of motor neurons toward more depolarizing values (Figure 1B). Of note, the time scale measured is comparable to that of cell division and differentiation, as previously reported [40].

### 2.2. Effects of PM0.1 and NP20 on Endocytosis, Mitochondrial Viability and ROS Production in NSC-34 Motor Neurons

Then, uptake of nanoparticles by motor-neuronal cells was determined by pHrodo™ Green, an intracellular dye emitting fluorescence in dependence of intravesicular pH. A significant increase in fluorescence emission within acidic vesicles was measured, starting from 3 h, in NSC-34 motor neurons exposed to both PM0.1 and NP20 (Figure 2A). Of note, brightfield images of motor neurons exposed to PM0.1 or NP20 for 3 h depicted round bodies reminiscent of endocytic vesicles (see Appendix A). The present data were expressed as Relative Fluorescence Units (RFU) measured up to 6 h and showed the occurrence of an impressive endocytosis.

Once their influx into the cells is verified, the effect of nanoparticles on mitochondrial bioenergetics has been studied by measuring several mitochondrial parameters in NSC-34 motor neurons exposed to PM0.1 and NP20 for different time points. Disruption of oxidative phosphorylation with a brief pulse of carbonyl cyanide 4-(trifluoromethoxy)phenylhydrazone (FCCP) demonstrated that neurons exposed to PM0.1 or NP20 for 12 and 24 h showed a time-dependent reduction in FCCP-induced [Ca^2+^]_i_ increase compared with untreated controls (Figure 2B(a–c)). This revealed a significant reduction in mitochondrial Ca^2+^ level in motor neurons exposed to nanoparticles. For both nanoparticles, these effects did not start before 12 h of exposure since no disruption of mitochondrial Ca^2+^ homeostasis was measured at 4 h of exposure (Appendix A). As subcellular correlate of oxidative phosphorylation dysfunction, ATP level was significantly reduced by chronic exposure to PM0.1 and NP20 (Figure 2C). Under these conditions, mitochondria of motor neurons exposed to PM0.1 and NP20 induced a detrimental release of cytochrome C (Figure 2C, inset) and ROS level generation (Figure 2D).

### 2.3. Effects of PM0.1 and NP20 on Lysosomal Channel TRPML1 and Autophagy in NSC-34 Motor Neurons

Being lysosomal, TRPML1 channels a specific sensor of ROS, required for lysosome adaptation to mitochondrial damage [41] and for the regulation of the autophagy flux [39,42], the effect of nanoparticles on TRPML1 expression and function has been investigated in motor neurons exposed to PM0.1 or NP20 for different time points.

Perfusion with the specific TRPML1 agonist, ML-SA1, induced a rapid [Ca^2+^]_i_ increase that was higher in Fura-2-loaded motor neurons previously exposed to PM0.1 or NP20 for 48 h (Figure 3A(a,b)). The same occurred in motor neurons exposed to nanoparticles for 24 h (data not shown). Moreover, the antioxidant molecule (±)-6-Hydroxy-2,5,7,8-tetramethylchromane-2-carboxylic acid (Trolox, 100 µM) significantly prevented the effect of nanoparticles on TRPML1 hyperactivation hampering ML-SA1-induced [Ca^2+^]_i_ increase (Figure 3A(a,b)). This was able to re-establish TRPML1 resting activity.

However, Western blotting analysis showed that chronic exposure to PM0.1 or NP20 triggered an increase in TRPML1, but not TPC2, channel expression (Figure 3B,C). The treatment with Trolox prevented TRPML1 protein expression increase (Appendix A).

In response to the deep increased activity of TRPML1, a variation in the autophagy markers has been detected in motor neurons exposed to PM0.1 or NP20. In fact, both LC3-II and p62 protein expression was increased after 48 h of exposure (Figure 4A(a–d)) together with AMPK activation (Figure 4B(a–c)) compared with resting expression under control conditions. These results suggested an engulfment or blockade of autophagy in motor neurons exposed to air pollution nanoparticles. The treatment with Trolox significantly reduced LC3-II protein expression in PM0.1-treated motor neurons, restoring the autophagy flux (Appendix A).

## 3. Discussion

The impact of air pollution on the central nervous system (CNS) is becoming even more relevant in industrial countries in which anthropogenic particulate matter (PM) is assuming a major detrimental role [10]. However, only PM components of a larger size (i.e., diameter = 2.5–10 µm) are currently monitored and covered by legal limits. On the other hand, the nanoparticle fraction of anthropogenic PM is not currently covered by legal limits since results are undetectable in the exhausts by the available methodologies [9]. Of note, PM toxicity appears to be inversely related to the size of PM fractions [10]. Although the historical focus of PM toxicity has been on the cardiopulmonary system [43], it is now obvious that inhaled nano-size particulates can enter circulation through the lungs and distribute throughout [44]. Molecularly, ultrafine PM particles can enter the brain through active transport, a blood–brain barrier leak, and a direct passage through the olfactory bulb [12], thus constituting a real noxious element triggering or potentiating neurodegeneration. However, only a few studies are available on this issue, thus rendering the knowledge on the interaction between ultrafine PM and CNS obscure.

In the present study, the neurotoxic effects of ultrafine PM fractions, PM0.1 and NP20, obtained by a properly developed analytical protocol, have been studied in motor-neuronal cells by analyzing their molecular mechanism and identifying a putative lysosomal druggable target against their neurotoxicity. Studying cell tracking by time-lapse and high-content microscopy, we showed that the smallest fraction of NP20 reduced NSC-34 motor neuron movement earlier in the space. Furthermore, NP20 was able to shift membrane potential of motor neurons toward more depolarizing values, as measured by the current-clamp method. This electrophysiological feature may impact the excitability of motor neurons, thus rendering them more vulnerable to eventual noxious stimuli [45].

Concomitantly, both of the ultrafine fractions PM0.1 and NP20 were able to enter into the cells by endocytosis and induced, intracellularly, detrimental effects on mitochondria and lysosomes. In this respect, PM0.1 and NP20 exerted mitochondrial toxicity with the consequent release of cytochrome C, decrease in ATP level, and stimulation of ROS production. This latter event was sufficient to determine the hyperactivation of the lysosomal channel TRPML1, which is considered a lysosomal ROS sensor [41]. Furthermore, the role of TRPML1 lysosomal channel in autophagy modulation is a consolidated acquisition even more relevantly highlighted by the specific activation of the transcription factor TFEB [39].

The CNS is particularly vulnerable to lysosomal dysfunction, causing impairment of neuronal function, which ultimately leads to neurodegeneration. For example, autophagosomes and lysosomes accumulate in the brain during several neurodegenerative diseases, thus highlighting the toxic effect of autophagy engulfment [34,37]. However, persistent activation of this cleansing mechanism can result in cell death, as occurs for environmental toxicants triggering autophagy in their target organs [46]. In our experiments, PM0.1 and NP20 induced an increase in LC3-II and p62 expression in motor neurons, thus determining autophagy engulfment. This was possibly linked to the ROS-dependent hyperactivation of the lysosomal channel TRPML1, whose persistence may be the cause of autophagy blockade during the chronic exposure. As proof of the fact that oxidative stress was involved, Trolox normalized TRPML1 function at a physiological level after exposure to the nano-size PM. Of note, the effect of Trolox on TRPML1 reverberated on downstream mechanisms, thus hampering autophagy flux, as demonstrated by the reduction in LC3-II expression. Therefore, it is possible that Trolox restored autophagy via TRPML1 function normalization. Most of the actions of the environmental contaminants, including PM at larger size, are mediated by ROS increase and AMPK activation [46]. Our study demonstrated that also nano-size PM engulfed autophagy by determining persistent activation of AMPK. Since defective mitochondria induce autophagy to promote cell survival through activating the AMPK pathway [47], we speculated that NP20- and PM0.1-induced AMPK phosphorylation may be considered as a protective mechanism.

Although representing a degradative pathway, autophagy serves as a protective mechanism during neurodegeneration [34,36,37,38]. Therefore, restoring the activity of the lysosomal TRPML1 at a physiological level may be considered a putative protective mechanism able to determine a protective resting autophagy flux. This highlights, once again, the role of lysosome as a hub of cellular homeostatic modulation introducing an additional role of lysosomal TRPML1 channel as a putative druggable target against environmental toxicants.

## 4. Materials and Methods

### 4.1. Ultrafine Particulate Isolation

Carbon PM was sampled in a lab-scale combustion system, as recently reported [43]. The set-up consisted of an atmospheric pressure, premixed ethylene/oxygen flame with an equivalence ratio, φ, equal to 1 and with a cold gas velocity equal to 4 cm/s, stabilized on a capillary burner on a water-cooled sintered bronze McKenna burner (d = 60 mm) (Holthuis & Associates, CA, Sebastopol, California USA).

PM was thermophoretically collected in this lab-scale combustion system inserting a glass plate horizontally into the flame by using a gear motor with a rotation speed of 1.4 gear/s.

The insertion time was set at 60 ms per lap with a total deposition time of 25 s. Care was taken to avoid the heating of the substrate and of the sample.

Carbon PM was scratched from the glass plates and extracted with dichloromethane (DCM) and then filtered for separating the soluble organic carbon (OC) from the DCM-insoluble ultrafine particles PM0.1.

PM0.1, collected on a Teflon filter, was suspended in dimethyl sulfoxide (DMSO) and further filtered on 20 nm pore size Anotop filters (Whatman), for separating particles <20 nm (NP20, constituting the 25% in weight of the total PM0.1 sample in the investigated experimental conditions) and particles >20 nm and <100 nm (NP100). Both fractions (NP20 and NP100) were dispersed in DMSO. In this study, we characterized the neurotoxicity of NP20.

More details on the flame properties and the sampling set up and separation procedure are reported in [9,48]. A detailed characterization of the OC and PM0.1 fractions has been reported in previous works. Briefly, OC is mainly composed of polycyclic aromatic hydrocarbons (PAHs) and high-molecular-weight aromatic species ([30] and references therein), whereas the PM0.1 fraction is extremely high in carbon with a low percentage of hydrogen (typically, the H/C atomic ratio is around 0.1 ([30] and references therein)).

### 4.2. Drugs and Chemicals

Media and reagents used for culturing cells were purchased from Gibco (ThermoFisher Scientific Inc., Waltham, MA, USA), and ECL reagents and nitrocellulose membranes were purchased from Cytiva Amersham (Amersham, UK). Reagents used for microfluorimetric experiments were purchased as follows: 2-(2-Oxo-2-(2,2,4-trimethyl-3,4-dihydroquinolin-1(2H)-yl)ethyl)isoindoline-1,3-dione (ML-SA1) from Merck Millipore (Darmstadt, Germany, DE); Carbonyl cyanide 4-(trifluoromethoxy)phenylhydrazone (FCCP), 2′,7′-dichlorofluorescein diacetate (DCFH-DA), 6-hydroxy-2,5,7,8-tetramethylchromane-2-carboxylic acid (Trolox), and the ATP bioluminescent assay kit from Sigma-Aldrich (Milan, IT); 1-[2-(5-Carboxyoxazol-2-yl)-6-aminobenzofuran-5-oxy]-2-(21-amino-51-methylphenoxy)-ethane-N,N,N1,N1-tetraacetic acid penta-acetoxymethyl ester (Fura-2/AM) was purchased from Molecular Probes (Invitrogen, ThermoFisher Scientific Inc., Waltham, MA, USA), DMSO (Merck Millipore, Burlington, MA, USA).

### 4.3. Cell Cultures and Treatments

Mouse motor-neuron-like cells (NSC-34) [49] were grown in Dulbecco’s Modified Eagle’s Medium (DMEM) supplemented with 10% fetal bovine serum (FBS), 2 mM L-glutamine, 100 IU/mL penicillin, and 100 μg/mL streptomycin. Cells were kept in a 5% CO_2_ and 95% air atmosphere at 37 °C. For experiments on single cells, NSC-34 motor neurons were plated on a glass coverslip coated with poly-L-lysine. Once adhered, cells were stimulated with PM0.1 and NP20 at the concentration of 2.86 ppm and 0.71 ppm, respectively. Control experiments were performed by adding DMSO (3%) in cell culture medium.

### 4.4. Time-Lapse and High-Content Microscopy

Microscopy experiments were conducted with the Operetta High-Content Imaging System (PerkinElmer, MA, USA), similar to previously described procedures [50,51]. Cells were cultured in Falcon^®^ 24-well Clear Flat Bottom. For time-lapse experiments, cells were stimulated with PM0.1 and NP20 and cultured for 6 h. Within this time window, digital phase contrast images of 15 fields/well were captured every 10 min via a 10× objective. To quantify cell tracking features, brightfield snapshots were taken at 6 fields/well. PhenoLOGIC (PerkinElmer) was employed to analyze kinetic proprieties as: current displacement X, current displacement Y, displacement X mean per well, displacement Y mean per well, displacement mean per well, and current speed.

### 4.5. Endocytosis Assay

NSC-34 motor neurons (1 × 10^5^ cells/well) were seeded into poly-L-lysine-coated imaging-compatible plates (BD, Falcon) 1 day before the assay. Once adhered (37 °C, 5% CO_2_), cells were stimulated with PM0.1 and NP20. Then, cell culture medium was removed and pHrodo™ Green *E. coli* BioParticles™ Conjugate suspension was added to the wells. pHrodo™ Green conjugates are nonfluorescent outside the cell at neutral pH but fluoresce brightly green at acidic pH such as in late endosomes. Therefore, the plate was placed in an EnSpire Multimode Plate Reader (Perkin Elmer). The data were expressed as Relative Fluorescence Units (RFU) measured up to 3 h with 1 h span at an excitation wavelength of 509 nm and emission at 533 nm.

### 4.6. Western Blotting

After treatments, cells were lysed through an ice-cold buffer containing 100 mM NaCl, 50 mM Tris-HCl (pH 7.4), 0.2% SDS, 0.5% NP-40, 1 mM EGTA, 1 mM PMSF, 1 mM Na_3_VO_4_, 1 mM NaF, and a protease inhibitor mixture (Roche Diagnostics, Monza, IT). For each sample, 30 μg of total proteins was separated by sodium dodecyl sulphate-polyacrylamide gel electrophoresis and then electrotransferred onto 0.2 μm nitrocellulose membranes. The membranes were blocked for 1 h in a bovine serum albumin-based buffer (3% BSA in TBS containing 0.1% Tween^®^20); primary antibodies were incubated overnight (see Table 1).

### 4.7. Electrophysiological Recordings

For spontaneous membrane potential recordings by current clamp, NSC-34 cells were bathed in extracellular Ringer solution containing (in mM): 126 NaCl, 1.2 NaHPO_4_, 2.4 KCl, 2.4 CaCl_2_, 1.2 MgCl_2_, 10 glucose, and 18 NaHCO_3_ at pH 7.4 (NaOH). The pipette solution contained (in mM): 140 K-Gluconate, 2 MgCl_2_, 2 Na_2_ATP, 0.3 NaGTP, 10 HEPES (pH 7.2 with KOH) [52,53,54]. The electrical activity was acquired in gap-free modality using a Digidata 1322A interface (Molecular Devices). Data were acquired and analyzed using the pClamp software (version 9.0, Molecular Devices). Electrophysiology data analysis was performed using Clampfit software (version 9.0, Molecular Devices). Spontaneous membrane potential was measured in NSC-34 cells under control conditions and in the presence of PM0.1 and NP20. Sustained high-quality whole-cell recordings could be maintained for >5 min confirming the viability of cells.

### 4.8. [Ca^2+^]_i_ Measurement by Single-Cell Video Imaging

NSC-34 motor neurons, plated on glass coverslips, were loaded with 10 µM Fura-2/AM at 37 °C. After 30 min, the coverslips with cells were mounted in a perfusion chamber (Medical System) posed onto an Axiovert200 microscope (Carl Zeiss) equipped with a FLUAR 40X oil objective lens and a Xenon lamp. The experiments were carried out with MicroMax 512BFT cooled CCD camera (Princeton Instruments), LAMBDA10–2 filter wheeler (Sutter Instruments), and Meta-Morph/MetaFluor Imaging System software (Universal Imaging). Fura-2/AM fluorescence intensity was measured every 3 s. Ratiometric values were automatically converted by the software to [Ca^2+^]_i_ using a calibration curve [55]. TRPML1 activity was studied by perfusing neurons with ML-SA1 (10 µM) in Krebs–Ringer saline solution (5.5 mM KCl, 160 mM NaCl, 1.2 mM MgCl_2_, 1.5 mM CaCl_2_, 10 mM glucose, and 10 mM HEPES-NaOH, pH 7.4) and calculated as percentage of [Ca^2+^]_i_ increase over basal values.

**Table 1 ijms-23-13041-t001:** List of primary antibodies used in the manuscript.

Antibody	Supplier	Catalog Number	Species	Type	Dilution Used	Application	Already Tested in:
β-actin-peroxidase	Sigma-Aldrich (Milan, Italy)	A3854 (RRID:AB_262011)	Mouse	Monoclonal	1:10000	WB	Tedeschi et al. [35]
Caspase 9 (cleaved Asp330)	GeneTex Inc. (Irvine, CA, USA)	GTX132331 (RRID:AB_2886615)	Rabbit	Polyclonal	1:1000	WB	Tedeschi et al. [37]
GRP78 (BiP)	Cell Signaling Technology, Inc. (Danvers, MA, USA)	3183 (RRID:AB_10695864)	Rabbit	Polyclonal	1:1000	WB	Secondo et al. [56] Tedeschi et al. [37]
LAMP1	Merck Millipore (Darmstadt, Germany)	AB2971 (RRID:AB_11212777)	Rabbit	Polyclonal	1:1000	WB	Tedeschi et al. [37]
					1:200	IP	
LAMP1	Santa Cruz Biotechnology, Inc. (Dallas, TX, USA)	sc-20011 (RRID:AB_626853)	Mouse	Monoclonal	1:1000	ICC	Tedeschi et al. [35]
LC3B	GeneTex Inc. (Irvine, CA, USA)	GTX127375 (RRID:AB_11176277)	Rabbit	Polyclonal	1:1000	WB	Tedeschi et al. [37]
p62/SQSTM1	Novus Biologicals (Littleton, CO, USA)	NBP1-48320 (RRID:AB_10011069)	Rabbit	Polyclonal	1:1000	WB	Tedeschi et al. [37]
TPC2	Alomone Labs (Jerusalem, Israel)	ACC-072 (RRID:AB_10918019)	Rabbit	Polyclonal	1:1000	WB	
					1:1000	ICC	
α-tubulin	Sigma-Aldrich (Milan, Italy)	T5168 (RRID:AB_477579)	Mouse	Monoclonal	1:5000	WB	Secondo et al. [56] Tedeschi et al. [35]

Applications. WB: Western blotting; IP: Immunoprecipitation; ICC: Immunocytochemistry.

### 4.9. Statistical Analysis

All the experiments were carried out in a blinded manner. Data were evaluated as means ± SEM. Statistically significant differences among means were determined by one-way ANOVA followed by Student–Newman–Keuls/Bonferroni post-hoc test. Student’s *t*-test was used for two groups comparison. Statistical significance was accepted at the 95% confidence level (*p* < 0.05). Statistical analyses were performed by using GraphPad Prism 5.0 (La Jolla, CA, USA).

## Figures and Tables

**Figure 1 ijms-23-13041-f001:**
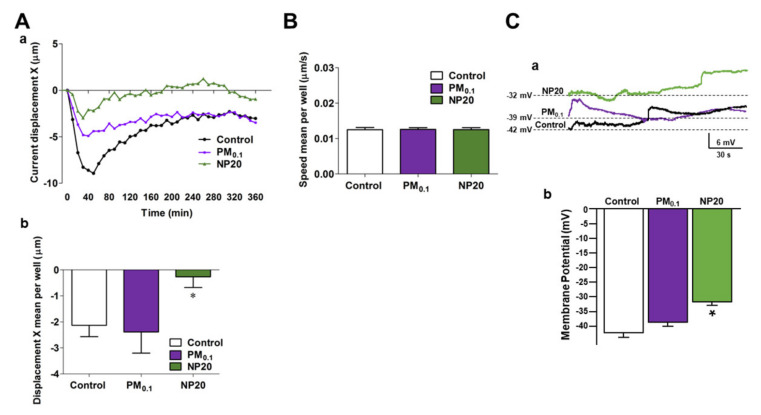
Effects of PM0.1 and NP20 on the kinetic and electrophysiological properties of NSC-34 motor neurons. (**A**,**B**) Tracking characteristics (current displacement X in Aa, displacement X mean per well in Ab and speed mean per well (**B**). NSC-34 (150 × 10^3^ cells/well) were stimulated (6 h, 37 °C) with DMEM alone (Control), PM0.1 (2.86 ppm) and NP20 (0.71 ppm). The incubation time was carried out in time-lapse and high-content microscopy Operetta High-Content Imaging System (PerkinElmer) per well. (**C**) Superimposed current-clamp traces (a) and relative quantification (b) of membrane potential. For each experimental group 5 cells have been used. * *p* < 0.05 vs. control and PM0.1.

**Figure 2 ijms-23-13041-f002:**
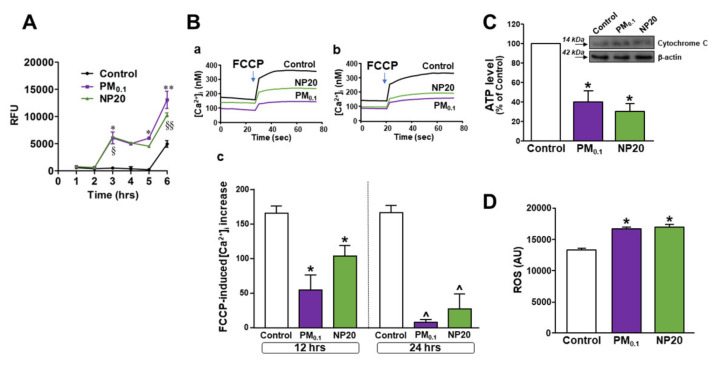
Effect of PM0.1 and NP20 on endocytosis and mitochondrial dysfunction in NSC-34 motor neurons. (**A**) Endocytosis in NSC-34 motor neurons calculated by pHrodo™ Green *E. coli* BioParticles™ Conjugate (37 °C, 5% CO_2_). The data were expressed as Relative Fluorescence Units (RFU) measured up to 6 h with 1 h span at an excitation wavelength of 509 nm and emission at 533 nm. NSC-34 (1 × 10^5^ cell/well) were stimulated (37 °C, 5% CO_2_) with PM0.1 (2.86 ppm), NP20 (0.71 ppm). * *p* < 0.05 vs. previous time points for the same treatment and same time point as control; § *p* < 0.05 vs. previous time points for the same treatment and same time point of Control; ** *p* < 0.05 vs. previous time points for the same treatment and control; §§ *p* < 0.05 vs. previous time points for the same treatment and control. (**B**) Superimposed traces representing the effect of FCCP on [Ca^2+^]_i_ after treatment with PM0.1 and NP20 for 12 h (a) or 24 h (b) and relative quantification as [Ca^2+^]_i_ increase (c). For each experiment at least n = 15 cells were detected. * *p* < 0.05 vs. its respective control at 12 h; ^ *p* < 0.05 vs. its respective control at 24 h. (**C**) Bar graph representing the level of ATP in NSC-34 motor neurons exposed to PM0.1 and NP20 for 48 h. * *p* < 0.05 vs. control. Western blot depicts cytochrome C released in cytoplasm after treatments. (**D**) Bar graph representing ROS level measured as arbitrary units in NSC-34 motor neurons exposed to PM0.1 and NP20 for 48 h. The experiments were repeated at least three times.

**Figure 3 ijms-23-13041-f003:**
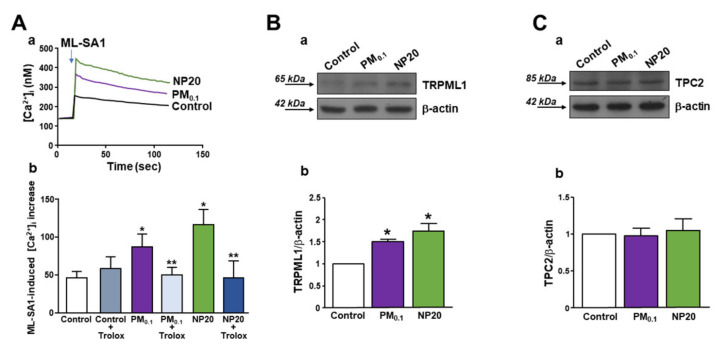
Effect of PM0.1 and NP20 on expression and activity of the lysosomal channel TRPML1. (**A**) Superimposed representative traces (a) and quantification (b) of the effect of ML-SA1 (10 µM) on [Ca^2+^]_i_ in NSC-34 motor neurons exposed to PM0.1 (2.86 ppm) and NP20 (0.71 ppm) (48 h) in the presence or absence of Trolox (100 µM). This latter drug was added every 24 h. All the experiments were repeated three times on almost 20 cells for Control, Control+Trolox, Control+PM0.1, Control +NP20 and 25 cells for PM0.1+Trolox and NP20+Trolox. * *p* < 0.05 vs. Control; ** *p* < 0.01 vs. PM0.1 or NP20. (**B**) Representative Western blotting of TRPML1 expression in NSC-34 motor neurons exposed to PM0.1 and NP20 (48 h) (a) and quantification (b). All the experiments were repeated three times on different preparations. * *p* < 0.05 vs. control. (**C**) Representative Western blotting of TPC2 expression in NSC-34 motor neurons exposed to PM0.1 and NP20 (48 hrs) (a) and quantification (b). All the experiments were repeated three times on different preparations.

**Figure 4 ijms-23-13041-f004:**
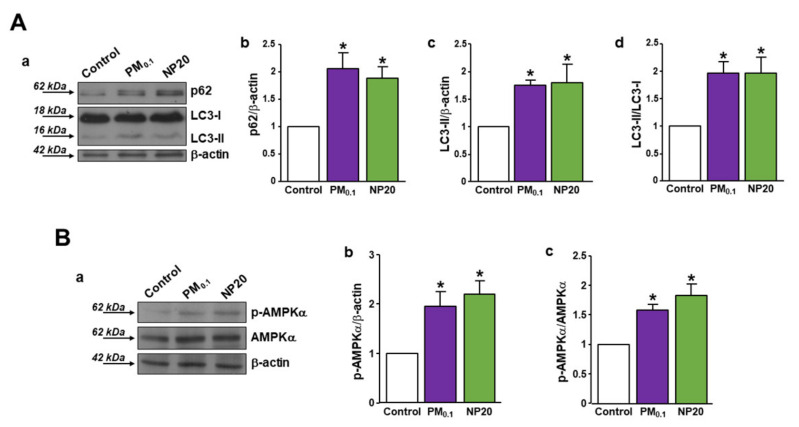
Effect of PM0.1 and NP20 on p62, LC3 and AMPK protein expression in NSC-34 motor neurons. (**A**) Representative Western blotting (a) and quantification (b–d) of p62 and LC3-II expression, respectively, in NSC-34 motor neurons exposed to PM0.1 (2.86 ppm) and NP20 (0.71 ppm) (48 h). Each bar represents the mean  ±  S.E. of data obtained from three different experimental sessions. * *p*  <  0.01 vs. each respective Control. (**B**) Representative Western blotting (a) and quantification (b,c) of p-AMPKα and AMPKα expression in NSC-34 motor neurons exposed to PM0.1 and NP20 (48 h). Each bar represents the mean  ±  S.E. of data obtained from three different experimental sessions. * *p*  <  0.01 vs. each respective Control.

## Data Availability

Not applicable.

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
