# Peer review of "Size-Based Effects of Anthropogenic Ultrafine Particles on Lysosomal TRPML1 Channel and Autophagy in Motoneuron-like Cells"

_ijms, 2022, doi:10.3390/ijms232113041_

Round 1

Reviewer 1 Report

Authors present in vitro study of the toxicity of ultrafine atmospheric particulate matter air pollutant particles with size below 100 nm (MP0.1) and 20 nm (NP20) on mouse hybrid neural cell line NSC-34.  The aim of the presented study was to investigate effect of endocytosed MP0.1 and NP20 environmental particles on lysosome dependent autophagy and mitochondrial functions, including correlation between nanoparticle-induced oxidative stress and Mucolipin 1 (TRPML1) dysfunction.

Major comments:

1.    The MP0.1 particles and NP20 particles are used at concentrations 2.86 ppm and 0.71 ppm, respectively. Authors conclude, based on the cited studies, that the in vitro system may provide evidence for in vivo mechanism of detrimental effect of such particles on human brain. What would be the expected concentration of air pollutant particles in human blood and/or in brain interstitial fluid?

2.    DMSO is used as a solvent/carrier for MP0.1 and NP20 particles. DMSO can affect mitochondrial processes and intracellular calcium ion concentration at relatively small concentrations (e.g. Verheijen, M. et al., Sci Rep 9, 4641, 2019, doi.org/10.1038/s41598-019-40660-0; Morley P. and Whitfield J. F., Journal of cellular physiology, 156, 219–225, 1993, doi.org/10.1002/jcp.1041560202). In Figure 1 the Control is declared as DMEM medium only. In other figures the information on Control used is missing. Authors should state, what the final concentration of DMSO in medium is, and whether the Controls used were medium only or DMSO at appropriate concentration. This information should be included either in methods or figure legends.

Minor comment:

In printed version of the manuscript some of the figure labels are too small to be visible.

Author Response

Question 1:

The MP0.1 particles and NP20 particles are used at concentrations 2.86 ppm and 0.71 ppm, respectively. Authors conclude, based on the cited studies, that the in vitro system may provide evidence for in vivo mechanism of detrimental effect of such particles on human brain. What would be the expected concentration of air pollutant particles in human blood and/or in brain interstitial fluid?

Answer:

This is an interesting and relevant question. However, at the moment, no study on NP quantification in the blood and SNC has been performed in human. This is due to the very low sensitivity to nanoparticles of the current commercial methods for detection. As a consequence, the current legal limits include limitations only up to PM2.5. Smaller particles are not covered yet by legal limits. The updated limits are reported in: https://www.breeze-technologies.de/blog/new-2021-who-air-quality-guideline-limits/.

Moreover, some interesting studies have been performed in preclinical animal models.  For instance mice exposed to ultrafine PM particles with concentrations about two times higher than ambient air in the some metropolitan areas, have a significant deposition of PM in lung and heart tissues with an estimated particle density of 3.0X107-8.0X107 particles for mm3 of each. This means that a high degree of circulation of these nanoparticles may occur in human. In this respect, it is relevant that the rate and degree of circulation of inhaled particles is considered as inversely correlated to the size.

In a very interesting paper by Tian group (Shang et al. NanoImpact 2021) an empirical equation has been developed to quantitatively predict the regional deposition rate of inhaled UFPs in human olfactory mucosa, a preferential route used to enter CNS. By applying this equation to the exposure in two different occupational scenarios, the authors stated that the continuous exposure to UFPs in humans determined a significant accumulation of particles in nasal olfactory tissue that seems to be of 1 to 4 times highest in workers  than in the counterparts not exposed.  In addition, this method could estimate the long-term brain accumulation of inhaled UFPs although its limitation due to underestimation of the real concentration of very low-size particles.

Question 2:

DMSO is used as a solvent/carrier for MP0.1 and NP20 particles. DMSO can affect mitochondrial processes and intracellular calcium ion concentration at relatively small concentrations (e.g. Verheijen, M. et al., Sci Rep 9, 4641, 2019, doi.org/10.1038/s41598-019-40660-0; Morley P. and Whitfield J. F., Journal of cellular physiology, 156, 219–225, 1993, doi.org/10.1002/jcp.1041560202). In Figure 1 the Control is declared as DMEM medium only. In other figures the information on Control used is missing. Authors should state, what the final concentration of DMSO in medium is, and whether the Controls used were medium only or DMSO at appropriate concentration. This information should be included either in methods or figure legends.

Answer

We thank the Referee for this important request. Of course we have performed controls for each type of experiment of the manuscript by adding DMSO in DMEM. In particular, DMSO was added up to 3% that is the maximum concentration used for NP experiments. At this concentration, no toxic effect was detected. This aspect was added in in the paragraph “cell culture and treatments” of Materials and Methods.

Question 3

Minor comment:

In printed version of the manuscript some of the figure labels are too small to be visible.

Answer

As requested, Figure labels has been increased

Reviewer 2 Report

It is worth adding more information about UFPM to the Introduction. UFPM in the atmosphere are unstable and subject to condensation and coagulation. These particles consist not only of elemental and organic carbon, but also sulfates, nitrates, trace elements, and adsorbed volatile organic compounds.

For fig. 1 A, B and 2 A, the resolution needs to be increased.

Author Response

Referee 2

Question 1:

It is worth adding more information about UFPM to the Introduction. UFPM in the atmosphere are unstable and subject to condensation and coagulation. These particles consist not only of elemental and organic carbon, but also sulfates, nitrates, trace elements, and adsorbed volatile organic compounds.

Answer:

We thank the Referee for this request that has been now resolved by adding much more details on ultrafine particulate matter in the Introduction section. Of note we substituted UFPM acronym with the right concept of ultrafine PM0.1 fraction (please see page 2 lines 58-79).

Question 2:

For fig. 1 A, B and 2 A, the resolution needs to be increased.

Answer:

As requested, the resolution of the Figures 1 and 2 has been implemented.